# Tear Proteomics in Children and Adolescents with Type 1 Diabetes: A Promising Approach to Biomarker Identification of Diabetes Pathogenesis and Complications

**DOI:** 10.3390/ijms25189994

**Published:** 2024-09-17

**Authors:** Eleni Angelopoulou, Rosa-Anna Kitani, Rafael Stroggilos, Vasiliki Lygirou, Ioannis-Anargyros Vasilakis, Konstantina Letsou, Antonia Vlahou, Jerome Zoidakis, Martina Samiotaki, Christina Kanaka-Gantenbein, Nicolas C. Nicolaides

**Affiliations:** 1Diabetes Center, Division of Endocrinology, Metabolism and Diabetes, First Department of Pediatrics, National and Kapodistrian University of Athens Medical School, “Aghia Sophia” Children’s Hospital, 11527 Athens, Greece; elen.angel@yahoo.gr (E.A.); vasilakisioan@yahoo.gr (I.-A.V.); chriskan@med.uoa.gr (C.K.-G.); 2Postgraduate Course of the Science of Stress and Health Promotion, School of Medicine, National and Kapodistrian University of Athens, 11527 Athens, Greece; rosakitani@gmail.com (R.-A.K.); dinaletsou@gmail.com (K.L.); 3Department of Biotechnology, Biomedical Research Foundation, Academy of Athens, 11527 Athens, Greece; rstrog@bioacademy.gr (R.S.); vlygirou@hotmail.com (V.L.); vlahoua@bioacademy.gr (A.V.); izoidakis@bioacademy.gr (J.Z.); 4Institute for Bio-Innovation, Biomedical Sciences Research Center “Alexander Fleming”, 16672 Vari, Greece; samiotaki@fleming.gr

**Keywords:** diabetic ketoacidosis, glycemic control, mass spectrometry, proteomics, tear proteome, type 1 diabetes

## Abstract

The aim of the current study was to investigate the tear proteome in children and adolescents with type 1 diabetes (T1D) compared to healthy controls, and to identify differences in the tear proteome of children with T1D depending on different characteristics of the disease. Fifty-six children with T1D at least one year after diagnosis, aged 6–17 years old, and fifty-six healthy age- and sex-matched controls were enrolled in this cross-sectional study. The proteomic analysis was based on liquid chromatography tandem mass spectrometry (LC-MS/MS) enabling the identification and quantification of the protein content via Data-Independent Acquisition by Neural Networks (DIA-NN). Data are available via ProteomeXchange with the identifier PXD052994. In total, 3302 proteins were identified from tear samples. Two hundred thirty-nine tear proteins were differentially expressed in children with T1D compared to healthy controls. Most of them were involved in the immune response, tissue homeostasis and inflammation. The presence of diabetic ketoacidosis at diagnosis and the level of glycemic control of children with T1D influenced the tear proteome. Tear proteomics analysis revealed a different proteome pattern in children with T1D compared to healthy controls offering insights on deregulated biological processes underlying the pathogenesis of T1D. Differences within the T1D group could unravel biomarkers for early detection of long-term complications of T1D.

## 1. Introduction

Type 1 diabetes (T1D) is a chronic autoimmune disorder characterized by progressive destruction of insulin-producing pancreatic beta cells, gradually resulting in insulin deficiency and hyperglycemia [1]. Soon after T1D diagnosis, the therapeutic target is to achieve optimal glycemic control through individualized exogenous insulin administration and regular glucose monitoring to prevent acute and long-term complications of the disease [2,3]. During the last few years, a growing body of evidence showed that the incidence of T1D diagnosis in early childhood is rising [4,5]. Interestingly, 42.3% of the newly diagnosed children in Sweden are aged 5–9 years, while 21.3% are diagnosed at an age younger than 4 years [4]. Early onset of the disease has been correlated both with higher frequency and severity of diabetic ketoacidosis (DKA) at diagnosis [6], and higher rates of neurocognitive dysfunction of affected children [7].

Mass spectrometry (MS)-based proteomics is a technology-driven science which allows the identification and quantification of a large number of proteins in various biological fluids, cells or tissues [8]. Proteomics contributes substantially to a deeper understanding of the pathophysiological mechanisms underlying diseases, and helps identifying protein biomarkers for early diagnosis, effectiveness of treatment and prognosis of an ever-increasing list of pathologic entities. In the field of diabetes, several proteomics studies in plasma, serum, saliva, urine, pancreatic samples and beta cells of affected subjects demonstrated significant differences in the expressed proteome, compared to healthy controls. Importantly, bioinformatics analyses showed that inflammation, the immune response and the complement activation cascade were among the most deregulated biological processes in T1D [9,10,11]. In addition, proteomics has been employed to unravel biomarkers for the early detection of diabetes complications, such as diabetic kidney disease and diabetic retinopathy, with promising results (reviewed in [12]).

Tears represent an easily collected biological fluid which plays a fundamental role in the defense of the eye against several pathogens. The tremendous progress of biological chemistry has enabled the characterization and systematic profiling of tear proteins, lipids and metabolites, paving the way towards identification of potential biomarkers for many ocular but also systemic diseases. Indeed, the tear film consists of an outer lipid layer, a median aqueous layer with large amounts of proteins, electrolytes and metabolites, and an inner layer with plenty of mucins. Tear proteins are mainly secreted from the lacrimal glands, whereas serum proteins are transferred to tears through the blood vessels, and some are produced by cells involved in the immune system processes present in the conjunctiva. Tear proteomics studies have been performed in several ocular diseases, such as dry eye disease and Sjogren’s syndrome, as well as in many systemic diseases, including Alzheimer’s disease and multiple sclerosis, yielding lists of differentially expressed proteins in patients compared to healthy subjects [13,14,15].

Tear proteomics in diabetes has been performed mainly in adults with type 2 diabetes, in an attempt to identify biomarkers for diabetic retinopathy [16]. To the best of our knowledge, tear proteomics has not been studied in children with T1D yet. The aim of this study is to investigate the tear proteomics profile in children and adolescents with T1D compared to healthy controls in an attempt to unravel potential biomarkers of the pathogenesis and of long-term complications of T1D. The identification of any differentially expressed proteins might provide further evidence for the autoimmune basis of T1D and the pathogenetic role of hyperglycemia-mediated long-term complications of the disease. Any potential biomarkers might be useful in clinical practice for optimization of glycemic control and prevention of diabetic complications.

## 2. Results

### 2.1. Clinical Characteristics and Laboratory Parameters of the Study Population

Children and adolescents with T1D enrolled in this study had a mean age (±SD) of 11.5 ± 2.4 years and healthy controls (HCs) a mean age (±SD) of 11.5 ± 2.5 years. The sex distribution of the participants in each group was 25 males and 31 females. The median duration of disease for children with diabetes was 2.2 (1.5–4.8) years. The clinical and laboratory data of all participants are presented in Table 1.

Within the diabetes group, further grouping was performed based on the presence of DKA at diagnosis and the level of children’s recent glycemic control. Regarding the occurrence of DKA at diagnosis, 25 children and adolescents experienced DKA at diagnosis whereas 29 did not (no available data for 2 participants). According to their metabolic control, 32 children with HbA1c > 56 mmol/mol (7.3%) were considered as the group with poor glycemic control, whereas 24 children with HbA1c ≤ 56 mmol/mol (7.3%) were considered as the group with good glycemic control. As expected, poor glycemic control is mostly observed in adolescents (advanced pubertal stage ≥ 3) compared to those in the prepubertal stage (*p*-value, 0.045) [17]. The clinical characteristics of participants with T1D are shown in Table 2. The clinical and laboratory data of each subgroup within the diabetes group are presented in Appendix A.

### 2.2. Identification of Proteins in Total Tear Samples

A total of 3302 proteins were identified from all tear samples, listed in Appendix A. Proteins that are known to be abundant in tears, including lysozyme C, lacritin, lipocalin 1 and lactotransferrin, were detected among the 794 proteins which were present in at least 90% of the tear samples, listed in Appendix A. Interestingly, the exosomal markers CD63, CD9, CD82 and tetraspanin-6 were detected in more than 60% of tear samples, whereas tetraspanin-14 and flotillin-1 were identified in less than 20% of the samples. Specifically, CD63 and CD9 were identified in all tear samples (Figure 1).

### 2.3. Differentially Expressed Proteins in Children and Adolescents with T1D Compared to Healthy Subjects

Two hundred thirty-nine tear proteins were differentially expressed in children and adolescents with T1D, compared to HCs. Proteins that displayed higher expression in the diabetes group are involved in biological processes of the adaptive immune response and tissue homeostasis. On the other hand, proteins with lower expression in the diabetes group participate in biological pathways of regulation of cellular component size, intracellular transport and cytoskeleton organization (Figure 2).

Among the proteins with statistically significant higher expression in tears of children with T1D compared to HCs are various proteins and immunoglobulins interfering in the adaptive immune response, such as IGLV4-60, IGLV4-69, IGLV3-12, IGLV7-46 and IGLV5-52 (Figure 3). Table 3 shows the twenty proteins with the highest fold change that exhibited statistical significance in the diabetes group compared to the HC group.

### 2.4. The Presence of DKA at Diagnosis Influences the Expressed Tear Proteome

Forty-four proteins were differentially expressed among children with T1D according to the presence or absence of DKA at diagnosis. These proteins are involved in biological processes of the response to toxic substances, the alpha amino acid catabolic process, and substantia nigra development. Some of the proteins, with the highest fold change that exhibited statistical significance in children who experienced an episode of DKA at diagnosis compared to those who did not, are presented in Table 4.

### 2.5. Participants with T1D and Good Glycemic Control Displayed a Distinct Tear Proteomics Profile, Compared to Those with Poor Glycemic Control

When comparing children with T1D depending on the level of their glycemic control—using as distinguishing cut-off value the levels of HbA1c≤ or >56 mmol/mol (7.3%)—120 proteins were found to be statistically differentially expressed between the two groups. Bioinformatics analysis revealed that the most deregulated biological processes in participants with poor glycemic control compared to those with good glycemic control are the acute phase response, adaptive immune response, immunoglobulin production and regulation of complement activation (Figure 4).

These processes were upregulated in participants with diabetes who do not achieve the desirable glycemic control. The 20 proteins with the highest fold change that exhibited statistical differences in tears of participants between these two groups are presented in Table 5.

## 3. Discussion

In the current study, the tear proteome of children and adolescents with T1D investigated at least 12 months after T1D diagnosis compared to age- and sex-matched healthy controls has been investigated. More than three thousand proteins were present in all tear samples, among which 794 proteins were present in 90% of the participants, including proteins that compose the core tear proteome, such as lysozyme C, lacritin, lipocalin 1 and lactotransferrin, confirming the quality of the samples. Proteomics analysis revealed that patients with T1D displayed a higher expression of proteins involved in the adaptive immune response, tissue homeostasis and defense response to bacteria, indicating increased immune response and inflammation processes. Indeed, several immunoglobulins and proteins participating in inflammation (e.g., trefoil factor 1 (TFF1), leucine-rich alpha-2-glycoprotein LRG1) were upregulated in tears of participants with T1D, compared to healthy controls, reinforcing the autoimmune and inflammatory pathogenetic origin of the disease. This is in line with results from an ever-increasing number of tear proteomics studies performed in patients with other autoimmune disorders, such as Sjögren’s syndrome [18], autoimmune thyroid disorders [19] and multiple sclerosis [20], suggesting that tear proteins might be used in clinical practice as biomarkers for diagnosis and follow-up of patients with autoimmune disorders. This clearly different tear proteome expression pattern in children with T1D investigated at least 12 months after diabetes diagnosis, could render the tear proteome analysis a promising approach in the elucidation of pathways implicated in diabetes pathogenesis.

In addition, significant alterations in the expression of proteins that interfere in neuronal homeostasis have been observed in children and adolescents with T1D in comparison to healthy controls. Agrin is a proteoglycan that plays an important role in the normal function of neurons, as well as in neuronal synapses [21]. ABCA1 (phospholipid-transporting ATPase ABCA1) is a phospholipid-transporting ATPase that is associated with Alzheimer’s disease [22]. Moreover, semaphorin 3E plays a fundamental role in neurodevelopment, angiogenesis and the formation of long axon tracts in the central nervous system [23]. Glutamate carboxypeptidase 2 is a metallopeptidase that contributes to glutamate homeostasis; importantly, high levels of this enzyme have been associated with neurocognitive disorders [24]. All these proteins were upregulated in tear samples of participants with diabetes, suggesting that chronic hyperglycemia and high variations in glucose levels might contribute to neuronal damage at the molecular and cellular level.

Moreover, within the diabetes cohort, when separately studying those who presented with DKA at diabetes diagnosis versus those who did not, it has been shown that those who experienced DKA had differential expression of proteins that interfere with the response to toxic substance, the alpha amino acid catabolic process and substantia nigra development. Children with DKA displayed a lower expression of proteins involved in response to toxic substances, such as peroxiredoxin 2 (PRDX2) and peroxiredoxin 6 (PRDX6). In addition, catabolic pathways, including that of alpha amino acids, have been found to be deregulated, as shown by the differentially expressed proteins homogentisate 1,2-dioxygenase (HGD), serine cytosolic hydroxymethyltransferase (SHMT1) and bleomycin hydrolase (BLMH). Importantly, a cluster of proteins involved in the development of the substantia nigra, including myelin basic protein (MBP), beta-arrestin-2 (ARRB2) and tubulin beta-4A chain (TUBB2A), were isolated in those participants who experienced DKA at diagnosis, possibly linking T1D with neurodegenerative diseases such as Parkinson’s disease in adulthood. Interestingly, Sanz and collaborators used Drosophila as a model of T1D to investigate whether T1D might represent a risk factor for future Parkinson’s disease occurrence [25]. The authors showed that Drosophila flies with increased soluble carbohydrate and glycogen levels presented with locomotor defects, and displayed decreased levels of tyrosine hydroxylase, a representative marker of dopaminergic neurons in the substantia nigra, as well as elevated markers of oxidative stress, also participating in the multifactorial pathogenesis of Parkinson’s disease [25,26]. There is a growing body of evidence in the international literature suggesting that the presence of DKA at diabetes diagnosis constitutes a risk factor for the future development of long-term diabetes complications [27]. In this context, there is an international effort to set the diagnosis of T1D at an earlier stage before the significant exhaustion of beta cell reserves. Interestingly, it has been demonstrated that DKA still influences the tear proteome in patients with T1D when investigated after 12 months from diagnosis, indicating the long-term consequences of the deregulation of the acid–base balance. Some of these differentially expressed proteins may be used as potential biomarkers for the long-term risk of future diabetes complications including cognitive impairment.

Furthermore, after subdividing the diabetes group into those with good glycemic control [HbA1c ≤ 56 mmol/mol (7.3%)] versus those with poor glycemic control [HbA1c > 56 mmol/mol (7.3%)], it has been demonstrated that the tear proteome of participants with poor glycemic control exhibits higher expression of proteins implicated in the acute phase response, immunoglobulin production and regulation of complement activation, thereby indicating a more exaggerated inflammatory pattern, suggesting that these differentially expressed proteins may be used as potential biomarkers for the impact of long-lasting hyperglycemia, including the future development of micro- or macrovascular diabetes complications. Several lines of evidence in the international literature show that chronic hyperglycemia due to inadequate glycemic control results in increased inflammation both in the periphery and the central nervous system (neuroinflammation) [28,29,30]. Indeed, exposure to prolonged hyperglycemia in poorly controlled patients with T1D has been associated with increased oxidative stress through overproduction of nicotinamide adeninedinucleotide (NADH) and mitochondrial reactive oxygen species (ROS) [31]. These molecules inhibit the glucose metabolism and trigger the activation of alternative glucose metabolic pathways, such as the hexosamine biosynthetic pathway and the polyol pathway, which both contribute to further ROS production. Moreover, chronic hyperglycemia activates several isoforms of protein kinase C that, in turn, leads to upregulation of several proinflammatory genes [32,33,34] and induces the production of advanced glycation end products (AGEs) [31,35]. These tissue-damaging compounds arise from non-enzymatic irreversible glycation of lipids, nucleic acids and proteins, and bind onto their cognate receptors for advanced glycation end-products (RAGEs). At the cellular and molecular level, RAGE signal transduction leads to (i) activation of NADPH oxidase, further causing mitochondrial DNA and protein damage [36]; (ii) activation of transcription factor nuclear factor-κB (NF-κB) which plays a fundamental role in the initiation and progression of inflammation through upregulation of cytokines, growth factors and RAGEs itself [37]; and (iii) the activation of mitogen-activated protein kinase leading to endothelial dysfunction, thereby illustrating the complex pathogenesis of diabetic vascular complications [38]. In addition to hyperglycemia-induced peripheral inflammation, similar molecular events occur in neuronal cells causing neuroinflammation [39] which contributes to neurocognitive dysfunction [7,40,41] and diabetes-induced dementia through downregulation of brain-derived neurotrophic factor (BDNF) [42]. Taking all these pathophysiological mechanisms of chronic hyperglycemia-mediated inflammation into consideration, we speculate that tear proteins identified in poorly controlled patients with T1D may contribute to inflammation through deregulation of biological processes, including the acute phase response, immune response and regulation of complement activation.

Importantly, exosomal markers have been identified in all tear samples, indicating that exosomes are present in tear samples containing molecules that might function as signal messengers, including DNA, RNAs and non-coding RNAs, peptides/proteins and/or metabolites. Exosomes have been identified and characterized in several autoimmune and non-autoimmune disorders [43,44]. Shi and collaborators investigated the context of exosomes isolated from tear samples of patients with Graves’ ophthalmopathy, patients with Graves’ disease without ophthalmopathy, and healthy controls. They demonstrated that the levels of cytokines IL-1 (interleukin-1) and IL-18 (interleukin 18), caspase-3, complement C4A and APOA-IV (apolipoprotein A-IV) were higher in tear exosomes of patients with Graves’ ophthalmopathy, compared to those without ophthalmopathy and healthy subjects [43]. In the current study, the levels of APOA-IV were found to be higher in children with T1D compared to HCs and the levels of complement C4A were found to be higher in children with diabetes presenting poor glycemic control compared to those with good glycemic control. Although we identified the exosomal markers CD63 and CD9 in all tear samples, we have not yet proceeded with their isolation and characterization in the context of tear exosomes. Nevertheless, it is likely that the proteins identified in the examined tear samples that are not included in the tear core proteome could constitute the cargoes of these exosomes, in this way transferring information from other cells and systems, such as central nervous system, into the tears.

To the best of our knowledge, this is the first study in the field of translational research investigating the tear proteome in children and adolescents with T1D in an attempt to timely identify potential biomarkers for both diabetes pathogenesis and possible future diabetes complications. The number of tear samples was relatively large for standard proteomics analysis and the study succeeded in identifying the largest number of proteins using the single-pot, solid-phase-enhanced sample preparation (SP3) protocol in combination with data-independent acquisition mass-spectrometry. The protein extraction buffer that has been used also acted as a lysis buffer since it contained SDS, effectively solubilizing all the Schirmer strip content.

However, there are some limitations of the current study. The sample of participants with diabetes and HbA1c < 53 mmol/mol (7%) that is mostly considered as the level of adequate glycemic control was relatively small; therefore, participants with HbA1c ≤ 56 mmol/mol (7.3%) were considered as those presenting good glycemic control in order to compare the two groups of adequate versus inadequate glycemic control. A further limitation of the study is the fact that the absolute concentrations of some of the differentially expressed proteins were not measured using ELISA. Finally, the validation of our results, concerning some of the identified proteins with interesting biological function, using an alternative methodology and a larger cohort could better delineate the clinical significance of these findings; however, the validation has not yet been performed due to funding restrictions and difficulties in obtaining tear samples in a larger cohort.

Whether the autoimmune and inflammatory pattern that was observed in the diabetes group tested at least 12 months after diabetes diagnosis is even more prevalent at diabetes onset remains to be confirmed. In addition, it is unknown whether other types of diabetes of non-autoimmune origin are associated with a similar pattern of differentially expressed proteins. Moreover, it is still unknown whether some of the neuronal proteins might be used as biomarkers of early recognition of patients with T1D at high risk for cognitive dysfunction and/or diabetic complications such as diabetic neuropathy. International efforts towards DKA prevention are of paramount importance to alleviate the burden of long-term diabetic complications and tear proteome biomarkers may be useful in identifying those patients at higher risk. In addition, the finding of exaggerated inflammation processes in poorly controlled patients with T1D compared to those with good glycemic control should further strengthen the efforts towards better glycemic control for all patients with diabetes.

In conclusion, in this cross-sectional study we identified differentially expressed proteins in tears of children and adolescents with T1D tested at least 12 months after diabetes onset, compared with healthy age- and sex-matched controls. Patients with T1D presenting with DKA at diagnosis have a significantly different tear proteomics profile after 12 months, compared to those who presented only hyperglycemia without DKA. Finally, it has also been shown that the quality of glycemic control influenced the tear proteome. Further studies are needed to confirm these results and unravel the role of these proteins as potential biomarkers of early identification of those at higher risk for future diabetic complications.

## 4. Materials and Methods

### 4.1. Study Design and Population

Fifty-six children and adolescents with T1D at least one year after diagnosis, aged 6–17 years old, that are followed at the Diabetes Center of the First Department of Pediatrics of the National and Kapodistrian University of Athens at “Aghia Sophia” Children’s Hospital in Athens, Greece, and fifty-six healthy age- and sex-matched controls, were enrolled in this cross-sectional study. Exclusion criteria included (i) comorbidities including other autoimmune disorders; (ii) diabetes complications; (iii) any abnormal findings in routine ocular examination within the previous year. The study protocol was conducted according to the guidelines of the Declaration of Helsinki and has been approved by the Scientific Committee of the “Aghia Sophia” Children’s Hospital (No 23645/24/11/2021). Written informed consent has been obtained from the parents or guardians of all participants following a detailed description of the aim and the design of the study before inclusion of the participants in the study.

### 4.2. Clinical and Laboratory Parameters

Medical history regarding diabetes diagnosis, including age of diagnosis, presence and degree (mild/moderate/severe) of DKA at diagnosis, HbA1c, C-peptide and T1D-related autoantibodies at diagnosis, as well as the history of any severe hypoglycemic episodes were recorded from the medical records of the children with diabetes. In addition, type of insulin treatment administration either with multiple daily injections (MDI) or insulin pump therapy as well as the use and data of flash (FGM) or continuous glucose monitoring systems (CGM), such as time in the target range (TIR), time above the target range (TAR), time below the target range (TBR) and coefficient of variation (CV), were also recorded. At the medical visit, weight (kg), height (cm), Body Mass Index (BMI) (kg/m^2^), blood pressure (mmHg), puberty status according to the Tanner stages and standard hematological and biochemical investigations including HbA1c, glucose and lipids, were assessed in all participants.

### 4.3. Assays

The standard hematologic parameters were assessed using the ADVIA 2110i analyzer (Roche Diagnostics, GmbH, Mannheim, Germany) while glucose, triglycerides, total cholesterol, HDL and LDL were quantified using the ADVIA 1800 Siemens analyzer (Siemens Healthcare Diagnostics, Tarrytown, NY, USA). HbA1c was measured using reversed-phase cation exchange high-performance liquid chromatography (HPLC) on an automated glycohemoglobin analyzer (HA-8160, Arkray, Kyoto, Japan).

### 4.4. Tear Sample Collection

Tear sampling was performed using Schirmer strips (BioSchirmer, biotech, Luzern, Switzerland, CE 2460) that were placed inside the lower eyelid of one eye, without anesthesia, for 5 min with closed eyes. Afterwards, the tear film was placed in a 1.5 mL sterile Eppendorf tube and was stored at −20 °C until proteomics analysis. Each Eppendorf tube was labeled pseudo-anonymously to protect personal data.

### 4.5. Tear Proteomics Protocol

The Schirmer strips were cut into pieces and submerged in a buffer consisting of 4% sodium dodecyl sulfate (SDS) (Sigma-Aldrich, St. Louis, MI, USA) and 0.1 M dithiothreitol (DTT) (Sigma-Aldrich) in 0.1M Triethylammonium bicarbonate buffer (SIGMA) (ΤΕΑΒ) and incubated for 5 min in 99 °C. Subsequently, the samples were subjected to sonication. The heating and sonication cycles were repeated twice. The protein extracts were processed by tryptic digestion using the Sp3 (Single-pot, solid-phase-enhanced sample preparation) protocol, including an alkylation step in 100 mM iodoacetamide (Acros Organics, Geel, Belgium) as Stergioti et al. described in detail [45]. The following day, the digest was purified using a modified Sp3 clean-up protocol for peptides using 95% acetonitrile when binding the peptides to the beads. The dried-down purified peptides were finally solubilized in mobile phase A (0.1% formic acid in water) assisted by sonication. The peptide concentration was determined measuring absorbance at 280 nm measurement using a nanodrop instrument [46].

Samples were analyzed on a liquid chromatography tandem mass spectrometry (LC-MS/MS) setup consisting of a Dionex Ultimate 3000 nanoRSLC coupled in line with a Q Exactive HF-X Orbitrap mass spectrometer (both from Thermo Fisher Scientific, Palo Alto, CA). The detailed protocol used was described by Stergioti et al. [45].

The generated DIA Orbitrap raw files were processed using the software DIA-NN (data-independent acquisition by neural networks) version 1.8.1. through searching against the *Homo sapiens* database (containing 20583 proteins downloaded from UniProt proteome database (https://www.uniprot.org/) accessed on 8 November 2022) using the library-free mode of the DIA-NN software (https://github.com/vdemichev/DiaNN, accessed on 8 November 2022), allowing up to two tryptic missed cleavages and a maximum of three variable modifications per peptide [47]. The double search mode of the software was utilized. A set of modifications was used in the search, including oxidation of methionine residues, N-terminal methionine excision and acetylation of the protein N-termini set as variable modifications and carbamidomethylation of cysteine residues as fixed modification. The match between runs feature was enabled for all analyses and the result was filtered at 0.01 FDR (false discover rate). Protein inference was performed on the level of genes using only proteotypic peptides.

The mass spectrometry proteomics data have been deposited to the ProteomeXchange Consortium via the PRIDE [48] partner repository with the dataset identifier PXD052994 (https://www.ebi.ac.uk/pride/, accessed on 8 November 2022).

### 4.6. Statistical Analyses and Graphs

The DIA-NN-quantified log2 protein intensities were loaded into the R environment [49] version 4.3, and the dataset was normalized to the natural logarithmic scale. We initialized the analysis by setting a strict frequency threshold of 10% on the missing values, meaning that across the 111 analyzed samples, only proteins having 11 or less missing values (n = 794 proteins) were kept, and we further named this subset of proteins as “tear core proteome”. Screening for the most prominent biomarkers was performed on the tear core proteome. In parallel, to increase the proteome coverage and explore other biologically relevant alterations, we performed a parallel analysis of the complete data, without applying any filtering steps. In both cases, proteins significantly different between the analyzed comparisons were determined with Mann–Whitney U tests (wilcox.test function) and significance was defined at an unadjusted *p* value < 0.05. Since sample sizes for some of the tested groups were relatively small (below 10 samples) adjustment of *p*-values for multiple comparisons would result in increased Type-II error and over-elimination of putatively important proteins, and thus was considered suboptimal. Benjamini–Hochberg-adjusted *p*-values are given in the results (Appendix A), but were not taken into consideration for any downstream filtering and analysis. Protein log_2_ fold change (natural logarithmic scale) for an example comparison X vs. Y was determined as the average log_2_ protein intensity in group X minus the average log_2_ protein intensity in group Y. Volcano plots were generated with the R libraries ggplot2 and ggrepel. The relevant R script created for both statistics and graphs, also containing information on the version of the used libraries, is given as Appendix A and can be quickly accessed with a text editor (e.g., notepad). Heatmap of the log_2_ protein intensities was created with the ComplexHeatmap package.

### 4.7. Pathway Enrichment

Pathway analysis was conducted on the web platform Metascape [50] (https://metascape.org/), accessed on 4 March 2022, individually for each of the comparisons, using the significantly (unadjusted *p* < 0.05) upregulated and downregulated proteins as input. The Gene Ontology—Biological Processes database was used for enrichment, and pathways having less than 3 genes were excluded from the analysis. The rest of the settings were kept at default and significantly enriched pathways were defined at Benjamini–Hochberg-adjusted *p*-values < 0.05.

## Figures and Tables

**Figure 1 ijms-25-09994-f001:**
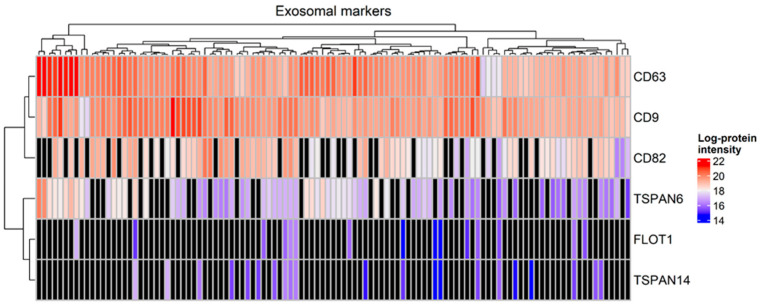
Heatmap with the expression of markers of exosomes in tear samples (Mann–Whitney test, *p*-value < 0.05). Black color denotes a missing value.

**Figure 2 ijms-25-09994-f002:**
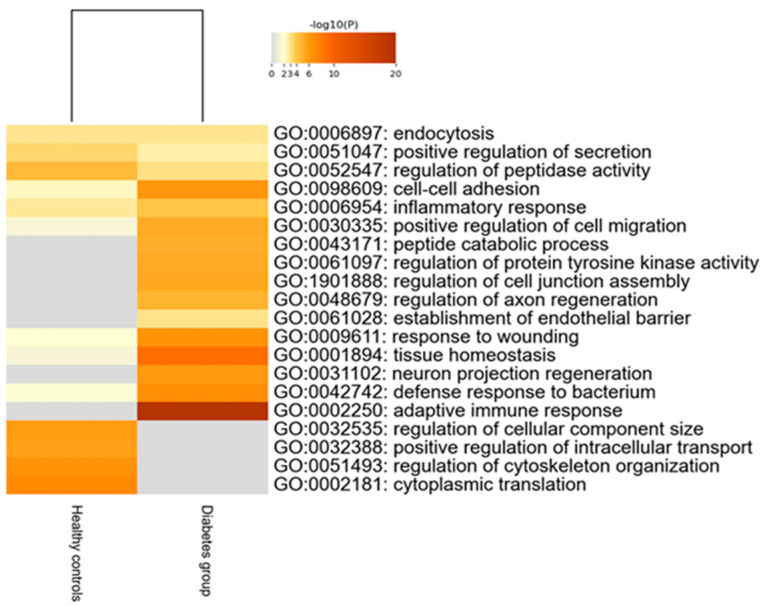
Heatmap of the deregulated biological pathways in children with type 1 diabetes compared to healthy controls (Mann–Whitney test, *p*-value < 0.05).

**Figure 3 ijms-25-09994-f003:**
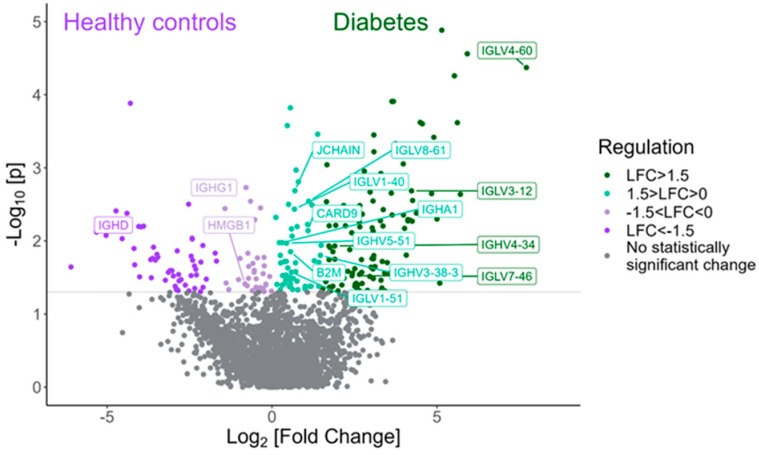
Volcano plot demonstrating the proteins involved in the adaptive immune response that were found to be statistically differentially expressed between the type 1 diabetes group and the HC group (Mann–Whitney test, *p*-value < 0.05).

**Figure 4 ijms-25-09994-f004:**
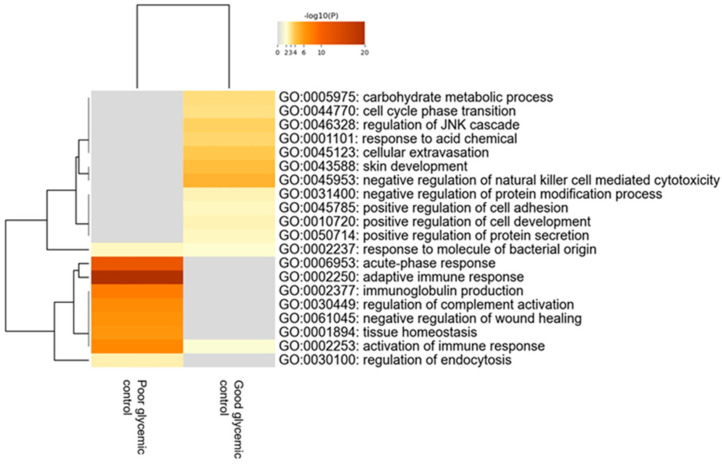
Heatmap of the deregulated biological pathways in children with type 1 diabetes with poor glycemic control [HbA1c > 56 mmol/mol (7.3%)] compared to those with good glycemic control [HbA1c ≤ 56 mmol/mol (7.3%)] (Mann–Whitney test, *p*-value < 0.05).

**Table 1 ijms-25-09994-t001:** Clinical characteristics and laboratory data of children with T1D and healthy controls.

	T1D Group (n = 56)	Control Group (n = 56)	*p*-Value
Mean age (years)	11.5 ± 2.4	11.5 ± 2.5	0.900
Sex (male/female)	25/31	25/31	1.000
Pubertal/ prepubertal status	37/19	32/21 (3 with no data)	0.557
Median BMI SDS	0.68 (−0.22, 1.47)	0.58 (−0.04, 1.80)	0.932
Median HbA1c (mmol/mol, %)	60 (54,66), 7.6% (7.1%, 8.2%)	36 (33, 37), 5.4% (5.2%, 5.5%)	**<0.001**
Median glucose (mg/dL)	170 (136, 200)	83 (80, 89)	**<0.001**
Median total cholesterol (mg/dL)	168 (144, 187)	155 (144,165)	**0.021**
Median LDL (mg/dL)	86 (70,111)	84 (72, 93)	0.324
Median triglycerides (mg/dL)	54 (42,64)	54 (47,74)	0.313

Values are expressed as mean (±SD) or median (25th and 75th percentiles) for continuous variables and as absolute numbers (n) and frequencies for categorical variables. Values in bold are statistically significant. T1D, type 1 diabetes.

**Table 2 ijms-25-09994-t002:** Clinical and laboratory data of children with T1D.

	T1D Group (n = 56)
Median disease duration (years)	2.2 (1.5, 4.8)
Mean age at T1D onset (years)	8.0 ± 3.3
Presence of DKA at diagnosis (yes/no)	25/29 (2 with no data)
No DKA/mild DKA/ moderate–severe DKA at diagnosis	29/13/12 (2 with no data)
Episodes of severe hypoglycemia (n)	3
FGM-CGM use/ no FGM-CGM use	45/11
Use of Insulin pump therapy/ multiple daily injections	7/49
Comorbidities	1 child with hypercholesterolemia

Values are expressed as mean (±SD) or median (25th and 75th percentiles) for continuous variables and as absolute numbers (n) and frequencies for categorical variables. CGM, continuous glucose monitoring; DKA, diabetic ketoacidosis; FGM, flash glucose monitoring; T1D, type 1 diabetes.

**Table 3 ijms-25-09994-t003:** Selected differentially expressed proteins between children with T1D and healthy controls.

Gene Name	Protein Name	Biological Function	*p*-Value	Log_2_(Fold Change) (T1D-HCs)
*TFF1*	Trefoil factor 1	Product of mucous epithelium, supporting its role as a physical barrier and expressed in inflammation occurring in various pathologies.	<0.001	8.61
*IGLV4-60*	Immunoglobulin lambda variable 4–60	Adaptive immunity.	<0.001	7.71
*AGRN*	Agrin	Proteoglycan, component of the basement membrane in different tissues. Central role in neuromuscular junction and undetermined role in brain synapses and neuron homeostasis.	<0.001	5.92
*ABCA1*	Phospholipid-transporting ATPase ABCA1	Transmembrane protein expressed in various tissues, essential for cholesterol homeostasis associated with neurological pathologies such as Alzheimer’s disease.	0.002	5.71
*LRG1*	Leucine-rich alpha-2-glycoprotein	Secreted glycoprotein, expressed during granulocyte differentiation and involved in cellular homeostasis, immunity, inflammation response and neovascularization.	0.0002	5.61
*LIPH*	Lipase member H	Hydrolase with a possible role in lipid metabolism. High expression is described in a variety of cancers.	<0.001	5.52
*SEMA3E*	Semaphorin 3E	Secreted protein involved in neurodevelopment and angiogenesis. Plays an important role in the formation of long axon tracts in the brain.	<0.001	5.14
*LORICRIN*	Loricrin	Major keratinocyte protein that contributes to the barrier function of the epidermal cornified cell envelope.	0.03	5.08
*FOLH1*	Glutamate carboxypeptidase 2	Metallopeptidase responsible for the conversion of NAAG to NAA and glutamate, resulting in increased levels of glutamate in the brain. Higher expression is associated with neurocognitive disorders.	0.005	5.00
*MAN2B1*	Lysosomal alpha-mannosidase	Lysosome protein implicated in the catabolism of N-linked carbohydrates and mainly expressed in the lung, pancreas, brain and leukocytes.	0.0006	4.95
*FAM110A*	Protein FAM110A	Located in the cytoplasm and involved in cell proliferation and differentiation. Its expression depends on the cell cycle and is higher when CD4 lymphocytes are stimulated.	0.02	−6.09
*TM9SF3*	Transmembrane 9 superfamily member 3	Predicted to be involved in protein localization to the membrane. Mainly unknown biological functions of TM9 family proteins (possible adhesion in immune response, tumor progression).	0.007	−5.32
*NOL12*	Nucleolar protein 12	Enables identical protein binding activity. Predicted to be active in nucleolus, regulating its structure. Possible association with aging.	<0.001	−5.29
*ODAD4*	Outer dynein arm-docking complex subunit 4	Localizes to ciliary axonemes and plays a role in the docking of the outer dynein arm to cilia. Mutations in this gene cause severely reduced ciliary motility and the disorder CILD35.	0.008	−5.03
*MFN1*	Mitofusin-1	Mitochondrial outer membrane GTPase that mediates mitochondrial clustering and fusion.	0.003	−4.7
*ACTN2*	Alpha-actinin-4; alpha-actinin-2	Actin binding, bundling protein linked to several forms of cardiomyopathy and myopathy.	0.009	−4.54
*SLC25A1*	Tricarboxylate transport protein, mitochondrial	Transport protein which regulates the movement of citrate across the inner membranes of the mitochondria.	0.004	−4.39
*MARK2*	Serine/threonine-protein kinase MARK2	Developmental protein involved in axon guidance, neuronal migration and activation of the Wnt signaling pathway.	0.0001	−4.29
*EFL1*	Elongation factor-like GTPase 1	Elongation factor implicated in protein biosynthesis and involved in Shwachman–Diamond syndrome 2.	0.01	−4.18
*TRIR*	Telomerase RNA component interacting RNase	Exonuclease activity, implicated in RNA binding.	0.006	−4.04

CILD35, ciliary dyskinesia; HCs: healthy controls; NAA: N-acetyl-aspartate; NAAG: N-acetyl-aspartyl-glutamate; T1D, type 1 diabetes; Wnt: wingless-related integration site pathway.

**Table 4 ijms-25-09994-t004:** Proteins differentially expressed between T1D children with DKA at diagnosis and those without DKA at diagnosis.

Gene Name	Protein Name	Biological Function	*p*-Value	Log_2_(Fold Change) (with DKA–without DKA)
*KIAA1217*	Sickle tail protein homolog	Developmental protein necessary for skeletal system development at the embryonic stage.	0.01	5.37
*PTK7*	Inactive tyrosine-protein kinase 7	Transmembrane receptor involved in multiple cellular processes regarding tissue homeostasis, and regulating the Wnt signaling pathway.	0.006	5.21
*FAM3C*	Protein FAM3C	Developmental protein involved in retinal laminar formation.	0.01	4.90
*INCENP*	Inner centromere protein	Part of the CPC, with essential function in mitosis and cell division.	0.03	4.80
*IRF3*	Interferon regulatory factor 3	Activator of IFN-dependent immune responses playing a role in immune processes against viruses.	0.02	4.52
*RP2*	Protein XRP2	GTPase activation with a role in protein transport between the Golgi apparatus and the ciliary membrane. It has been associated with retinitis pigmentosa.	0.03	4.31
*ARRB2*	Beta-arrestin-2	Signal transduction inhibitor with high expression in the brain.	0.02	4.29
*ERAP2*	Endoplasmic reticulum aminopeptidase 2	Aminopeptidase involved in appropriate antigen presentation by MHC class I molecules and playing a role in immune processes.	0.007	4.21
*PRH2*	Salivary acidic proline-rich phosphoprotein 1/2	Mainly salivary protein that acts as inhibitor of crystal growth of calcium phosphates.	0.02	4.14
*CPN2*	Carboxypeptidase N subunit 2	Located in blood microparticles and extracellular exosomes and involved in protein stabilization.	0.01	4.09
*AKR1E2*	1,5-anhydro-D-fructose reductase	NADPH-dependent reductase which, in animal experiments, prevents the formation of AGEs.	0.01	−7.20
*MYH11*	Myosin 7B, myosin-9	Actin-binding muscle protein that plays a role in muscle contraction through ATP hydrolysis.	0.02	−5.29
*AGT*	Angiotensinogen	Protein with vasoactive molecular functions, that constitutes an essential component of the renin–angiotensin system (RAS).	0.02	−4.86
*ACTN2*	Alpha-actinin-4, alpha-actinin-2	Bundling protein with actin-binding molecular functions in different structures in the cells.	0.03	−4.84
*ATP1A3*	Sodium/potassium-transporting ATPase subunit alpha-1, sodium/potassium-transporting ATPase subunit alpha-3	ATPase responsible for the ion transport through the plasma membrane, with high expression in the brain.	0.02	−4.37
*CKB*	Creatine kinase B-type	Transferase involved in energy homeostasis.	0.009	−4.33
*HGD*	Homogentisate 1,2-dioxygenase	Enzyme involved in the catabolism of tyrosine and phenylalanine and related to alkaptonuria.	0.04	−4.27
*CNP*	2′,3′-cyclic-nucleotide 3′-phosphodiesterase	One of the most abundant proteins in the central nervous system’s myelin that is implicated in neurodegeneration.	0.02	−4.23
*GSDMB*	Gasdermin-B	Precursor of a pore forming protein involved in cytolysis and pyroptosis.	0.01	−3.27
*DDAH1*	N(G)-dimethylarginine dimethylaminohydrolase 1	Hydrolase playing a role in the regulation of nitric oxide production.	0.04	−2.57

CPC, chromosomal passenger complex; MHC, major histocompatibility complex; RAS, renin–angiotensin system; T1D, type 1 diabetes.

**Table 5 ijms-25-09994-t005:** Proteins differentially expressed between T1D children with HbA1c > 56 mmol/mol (7.3%) and those with HbA1c ≤ 56 mmol/mol (7.3%).

Gene Name	Protein Name	Biological Function	*p*-Value	Log_2_(Fold Change) (Poor Glycemic Control–Good Glycemic Control)
*MCM6*	DNA replication licensing factor MCM6	DNA Hydrolase implicated in DNA replication and cell cycle.	0.001	7.89
*AKR1E2*	1,5-anhydro-D-fructose reductase	NADPH-dependent reductase which, in animal experiments, prevents the formation of AGEs.	0.02	7.74
*FAM110A*	Protein FAM110A	Located in the cytoplasm and involved in cell proliferation and differentiation. Its expression depends on the cell cycle and is higher when CD4 lymphocytes are stimulated.	0.02	6.70
*RPLP0*	60S acidic ribosomal protein P0	Ribosomal protein implicated in protein synthesis with a possible role in cellular apoptosis.	0.01	6.31
*COL18A1*	Collagen alpha-1 (XVIII) chain	Secreted proteoglycan that probably plays a major role in determining the retinal structure as well as in the closure of the neural tube. Its C-terminal fragment, endostatin, has antiangiogenic functions.	0.01	4.88
*C4B_2*	Complement C4-A; complement C4-B	Non-enzymatic component of the C3 and C5 convertases involved in the activation of the classical complement pathway.	0.01	4.86
*HPR*	Haptoglobin; haptoglobin-related protein	Secreted acute phase plasma protein with antioxidant and anti-inflammatory effects.	0.01	4.69
*SURF4*	Surfeit locus protein 4	A cargo receptor regulating the export of proteins, mainly lipoproteins, to the Golgi system.	0.01	4.48
*RPL23A*	60S ribosomal protein L23a	Ribosomal protein having a structural activity in the cell.	0.03	4.38
*CASP6*	Caspase-6	Protease implicated in apoptosis and programmed cell death, with a role also in axonal degeneration and the immune response.	0.04	4.13
*GSDMA*	Gasdermin-A	Pore-forming protein involved in pyroptosis.	0.001	−7.21
*CALML5*	Calmodulin-like protein 5	Calcium-binding protein involved in the differentiation of keratinocytes.	0.003	−6.96
*EHD4*	EH domain containing protein 4	ATP- and membrane-binding protein that regulates membrane reorganization.	0.03	−6.64
*UPF1*	Regulator of nonsense transcripts 1	Part of a multiprotein complex involved in mRNA nuclear export and mRNA surveillance.	0.015	−6.46
*CRK*	Adapter molecule crk	Proto-oncogene involved in several signaling pathways playing a role in cell adhesion and branching.	0.004	−6.29
*MYD88*	Myeloid differentiation primary response protein MyD88	Cytosolic adapter protein involved in inflammatory processes and the immune response.	0.01	−6.18
*LCMT1*	Leucine carboxyl methyltransferase 1	Enzyme which, in an opposing way, together with PPME1 (protein phosphatase methylesterase 1) regulates the methylation of protein PPA2 that is expressed in the brain and implicated in neuronal signal transduction.	0.009	−6.17
*PPME1*	Protein phosphatase methylesterase 1	Enzyme which, in an opposing way, with LCMT1 (leucine carboxyl methyltransferase 1) regulates the methylation of protein PPA2 that is expressed in the brain and implicated in neuronal signal transduction.	0.005	−5.98
*COL1A2*	Collagen alpha-2 (I) chain	Pro-alpha2 chain of type I collagen playing a role in diseases such as the Ehler–Danlos syndrome and osteogenesis imperfecta.	0.005	−5.91
*CARS1*	Cysteine–tRNA ligase, cytoplasmic	Aminoacyl-tRNA synthetase involved in protein biosynthesis in the cytoplasm, abundant in the exocrine pancreas and neuronal cells.	0.04	−5.79

C3, complement C3; C5, complement C5; PPA2, inorganic pyrophosphatase 2, mitochondrial; T1D, type 1 diabetes.

## Data Availability

The mass spectrometry proteomics data have been deposited to the ProteomeXchange Consortium via the PRIDE [48] partner repository with the dataset identifier PXD052994.

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
