# Peer review of "Tear Proteomics in Children and Adolescents with Type 1 Diabetes: A Promising Approach to Biomarker Identification of Diabetes Pathogenesis and Complications"

_ijms, 2024, doi:10.3390/ijms25189994_

Round 1

Reviewer 1 Report

Comments and Suggestions for Authors

The Manuscript is fluent and the idea behind the work is very nice. The idea of the tears as potential source for T1 biomarkers is very interesting, but the aim of the whole work is not adequately eviscerated and required major revisions. The work lacks of an experimental validation or a structured statistical analysis in order to assess the associations between enriched pathways results and biological variables. 

Comments on the Quality of English Language

Minor editing

Author Response

Comments and Suggestions for Authors: The Manuscript is fluent and the idea behind the work is very nice. The idea of the tears as potential source for T1 biomarkers is very interesting, but the aim of the whole work is not adequately eviscerated and required major revisions. The work lacks of an experimental validation or a structured statistical analysis in order to assess the associations between enriched pathways results and biological variables.

Response: We thank the Reviewer for his/her comment. We have added a short part of the text explaining in more details the aim of our study (page 2, highlighted in yellow color). As far as experimental validation is concerned, we have stated that one limitation of our study was that we did not perform ELISA to measure the absolute concentrations of some of the differentially expressed proteins (page 14, highlighted in yellow color). Regarding statistical analysis, we performed standard statistical comparisons and utilized both strict and relaxed thresholds for missing values. As reported in the main text (page 16, highlighted in yellow color), we initialized the analysis by setting a strict frequency threshold of 10% on the missing values. This actually means that across the 111 analyzed samples, we kept only proteins having 11 or less missing values (n = 794 proteins), and we further named this subset of proteins as “tear core proteome”. We consider this as the most reliable subset to screen for biomarkers. However, since our study has a strong exploratory component, to increase the coverage of the proteome and potentially capture other related alterations, we decided to analyze in parallel the entire data as it is. Of course this comes with limitations for proteins with many missing values, or for comparisons with small group sample sizes and we are fully aware of them, but it provides a broader spectrum of the potential alterations for the curious reader. In the main text we placed increased emphasis on the core proteome alterations, and we acknowledged the limited sample sizes.

Reviewer 2 Report

Comments and Suggestions for Authors

see attached file

Author Response

The aim of the presented study was to detect changes in the tear proteome in children and adolescents with type 1 diabetes compared to healthy controls. The proteomic analysis of 56 patients and 56 healthy controls was performed on LC-MS/MS system enabling the identification and quantification of the protein content via DIA.

The design of the experiment as well as the bioinformatic processing of the results is at a high level and I have only a few comments/questions on the article.

Comment 1: Please do not give vagueness like most of tear samples (line 118) or large number of protein (line 180) instead use an exact number or statement like: more than 95% samples, more than 2000 identified proteins etc.

Response 1: Amended as suggested by the Reviewer (page 4, highlighted in yellow color; page 12, highlighted in yellow color).

Comment 2: Tables 3,4,5 take up too much space. Please consider redesigning them if possible.

Response 2: We agree with the Reviewer that Tables 3, 4 and 5 are too long; however, we have included all the relevant information about the biological function of each protein to show the possible involvement in biological processes. We have now slightly redesigned these Tables.

Comment 3: When analyzing the enrichment of signaling pathways within the diabetes group divided into subgroups DKA and no DKA, respectively HbA1c> 56 mmol/mol and with HbA1c≤ 56 mmol/mol, the authors quantified and took into account or PEA all measured proteins, regardless of the number of identifications of a specific protein within the samples subgroups. In the most extreme cases, it was protein quantification that was found in 0/25 DKA vs 5/29 no DKA subgroup. Wouldn't it be more appropriate to choose a threshold of, for example, 50% of the occurrence of a given protein in all measured samples for the following PEA?

Response 3: We thank the reviewer for pointing out this very important aspect of the analysis. Proteins with high frequency of missing values are usually filtered out, and this filtering is always decided by arbitrary thresholds. As reported in the main text (page 16, highlighted in yellow color), we initialized the analysis by setting a strict frequency threshold of 10% on the missing values. This actually means that across the 111 analyzed samples, we kept only proteins having 11 or less missing values (n = 794 proteins), and we further named this subset of proteins as “tear core proteome”. We have considered this as the most reliable subset to screen for biomarkers. However, since our study has a strong exploratory component, to increase the coverage of the proteome and potentially capture other related alterations, we decided to analyze in parallel the entire data as it is. Of course this comes with limitations for proteins with many missing values, and we are fully aware of this limitation, but it provides a broader spectrum of the potential alterations for the curious reader. In the main text we placed increased emphasis on the core proteome alterations, and we acknowledged the limited sample sizes.

Comment 4: Figure 1 shows the Heatmap with the expression of markers of exosomes in tear samples (Mann-Whitney test, 121 p-value<0.05 but the legend does not show the full scale of log protein intensity (0-22) but only 14-22. I assume that the black color means that the given protein was not identified. In this case, the proteins FLOT1 and TSAPN14 were not identified in most of samples as stated by the authors of the study (line 118) but in less than 20%.

Response 4: We thank the Reviewer for this comment. We have amended the manuscript stating that Tetraspanin-14 and Flotillin-1 were identified in less than 20% of the samples (page 4, highlighted in yellow color). We have also stated that “Black color denotes a missing value” (page 4, figure 4 legend, highlighted in yellow color).

Round 2

Reviewer 1 Report

Comments and Suggestions for Authors

Dear authors,

the revisions are not adequately completed. 

In this form the paper is only descriptive, although the originality of the work.

I suggest to perform at least a separate validation of the most frequent proteins (at least 2) reported in your analysis. 

Author Response

Dear Editor in Chief,

Dear Reviewer,

First of all, we would like to thank you again for the time and effort you have put in reviewing and providing constructive remarks to our revised manuscript.

Please find below a reply to the reviewer’s comment:

We understand the concern about the missing validation of our results using an alternative methodology and a larger cohort (page 14, highlighted in yellow color). However, we still consider that reporting this new proteomics approach by tear proteomics analysis is novel and worthy to be reported, both to demonstrate an alternative valuable biological fluid in the identification of biomarkers and also to set the basis for future research on this field.

The inability at this stage to perform the validation of our results is originating from the difficulty of collecting new pediatric tear samples both in respect of time and funding.

We have included this limitation in the “limitations of the study” part of the discussion in the revised manuscript highlighted in yellow.

We aim to continue this research as our results, although only descriptive, have been very interesting, providing a very deep new dataset on the tear proteome, and we plan to seek new funding for this. Unfortunately, this will not be realistic in the near future.

We thank you once again for your fruitful suggestions.

On behalf of all co-authors